# Optimization of Heparin Monitoring with Anti-FXa Assays and the Impact of Dextran Sulfate for Measuring All Drug Activity

**DOI:** 10.3390/biomedicines9060700

**Published:** 2021-06-21

**Authors:** Jean Amiral, Cédric Amiral, Claire Dunois

**Affiliations:** 1SH-Consulting, 78570 Andresy, France; 2HYPHEN BioMed, 95000 Neuville sur Oise, France; camiral@hyphen-biomed.com (C.A.); cdunois@hyphen-biomed.com (C.D.)

**Keywords:** heparins, anti-FXa assays, automation, calibration curves superimposition, dextran sulfate

## Abstract

Heparins, unfractionated or low molecular weight, are permanently in the spotlight of both clinical indications and laboratory monitoring. An accurate drug dosage is necessary for an efficient and safe therapy. The one-stage kinetic anti-FXa assays are the most widely and universally used with full automation for large series, without needing exogenous antithrombin. The WHO International Standards are available for UFH and LMWH, but external quality assessment surveys still report a high inter-assay variability. This heterogeneity results from the following: assay formulation, designed without or with dextran sulfate to measure all heparin in blood circulation; calibrators for testing UFH or LMWH with the same curve; and automation parameters. In this study, various factors which impact heparin measurements are reviewed, and we share our experience to optimize assays for testing all heparin anticoagulant activities in plasma. Evidence is provided on the usefulness of low molecular weight dextran sulfate to completely mobilize all of the drug present in blood circulation. Other key factors concern the adjustment of assay conditions to obtain fully superimposable calibration curves for UFH and LMWH, calibrators’ formulations, and automation parameters. In this study, we illustrate the performances of different anti-FXa assays used for testing heparin on UFH or LMWH treated patients’ plasmas and obtained using citrate or CTAD anticoagulants. Comparable results are obtained only when the CTAD anticoagulant is used. Using citrate as an anticoagulant, UFH is underestimated in the absence of dextran sulfate. Heparin calibrators, adjustment of automation parameters, and data treatment contribute to other smaller differences.

## 1. Background and Introduction

Heparin and its derivatives, including unfractionated heparin (UFH), low molecular weight heparin (LMWH), and fondaparinux, comprise a major group of anticoagulants with multiple indications in various clinical situations associated with occurrence or prevention of thrombosis [1,2,3]. Since its discovery [4,5], heparin has been used for the treatment of thrombotic diseases, and this has reversed their prognosis [6,7,8,9]. Therapy monitoring is required for drug dosage adjustment [10,11], especially when clearance is impaired [12,13], or in the presence of heparin resistance [14,15,16,17]. Rare and severe side effects can develop such as heparin-induced thrombocytopenia (HIT) [18,19]; however, heparin remains to be the anticoagulant of choice in many critical circumstances [1,8,20,21,22] and has anticoagulant activity through additional mechanisms [23,24,25]. UFH and LMWH drug dosages need to be accurately adjusted for each treated patient [2,6,13,26,27]. Inappropriate drug adjustment can generate severe and life-threatening thrombotic or bleeding complications. In addition, measurement accuracy is important for therapy monitoring. Many assays have been developed over time for monitoring heparin [11,28,29], such as the activated clotting time (ACT) used in cardiology [28,29,30] and the activated partial thromboplastin time (APTT) performed on citrated plasma [31,32]. APTT is still the first line method in many countries, despite its limitations [33,34]. The availability of chromogenic assays, designed with thrombin or FXa specific substrates, has permitted the development of more specific methods for testing heparin [10,30,33,35,36,37,38]. This has led to the development, first, of two-stage assays, and then anti-FXa kinetic automated techniques, which are the most widely used [33,35,38], with little interferences [31]. It is important to note that heparin is an indirect catalytic inhibitor and requires AT for inhibiting coagulation serine esterases [39,40,41,42,43]. For one-stage kinetic assays, no exogenous AT is needed, but the endogenous plasma AT must be ≥50% to avoid underestimating drug concentration. Heparin can interact with many blood proteins or blood cells through exposed surface proteins [44,45,46]. These proteins, such as platelet factor 4 (PF4) or plasma proteins such as histidine-rich glycoprotein (HRGP), bind and neutralize heparin [4,47,48], which can underestimate measured concentrations. The assays used must be those that are less sensitive to this limitation.

For the two types of chromogenic assays available, the two-stage method and the one-stage kinetic test [49,50,51], heparin calibrators obtained by spiking heparin in platelet-depleted plasma are required. Automated one-stage kinetic anti-FXa methods are currently used for monitoring heparin therapy, in association with UFH, LMWH, or fondaparinux plasma calibrators [30,52]. The WHO International Standards (IS) are available for UFH and LMWH and allow standardization and traceability of calibrators [53,54]. Although a like-to-like calibration is recommended for testing heparins, there is a strong market requesting a single calibration for UFH or LMWH, and it is now proposed by most manufacturers.

Despite the many efforts for anti-FXa assay standardization, many differences are still observed for heparin measurements when the various branded chromogenic methods are used [55,56,57,58]. This is illustrated by the external quality assessment programs, such as ECAT, showing a significant reagent-to-reagent and laboratory-to-laboratory variability, especially for UFH in the low range [57]. Recently, with the extended indications of UFH or LWMH treatments in COVID-19 patients, the debate on anti-FXa exactness and accuracy has been reactivated [55,57,59,60].

The design of anti-FXa assay conditions is essential for their performances. Many years ago, the use of low molecular weight dextran sulfate (DS) was introduced for improving the recovery of heparin measurements and avoiding the impact of neutralizing proteins [61,62,63]. This component is now used for most anti-FXa assays, but not for all of them [57]. In addition, using a single calibration for all heparin types, i.e., UFH or LMWH, is an important laboratory requirement; heparin is often tested in emergency conditions and the heparin type is not always known. This is possible when assay conditions are optimized to permit obtaining the same dose–response curve for UFH and LMWH [61,62]. However, some manufacturers prefer to propose a hybrid curve by mixing or combining UFH and LMWH to get a median curve [64].

To understand the differences between anti-FXa assays, reagents and reference material from various manufacturers have been compared for measuring plasmas from UFH or LMWH treated patients, anticoagulated with citrate or citrate-theophylline-adenosine-dipyridamole (CTAD), an anticoagulant developed for preventing platelet activation ex vivo, and the release of heparin neutralization proteins [65]. The differences among various assays were recently analyzed by Hollestelle et al., in a retrospective and multicentric study, using heparin spiked samples, tested at several sites participating in the external quality assessment program from ECAT [57]. Our report concerns a monocentric study, performed with plasmas from UFH or LMWH treated patients, and prepared from blood collected with either citrate or CTAD. In this study, we confirm the role of dextran sulfate for the complete measurement of all heparin activity, especially for the low UFH concentration range and when citrate is used as an anticoagulant, and we analyze the impact of the calibrators’ designs and formulations.

## 2. Materials and Methods

### 2.1. Patients and Normal Plasmas

Normal citrated plasmas and plasma pools were supplied frozen by Precision Biologic Inc. (Halifax, Canada), and stored at < −70 °C until use. Plasmas from hospitalized patients treated with heparin for post-surgery thrombosis prevention, with either UFH or LMWH, were obtained from Beaujon University Hospital (Clichy, France), as the left-over residual plasma from an ongoing clinical study and obtained in agreement with CLSI. According to the primary study protocol, blood was collected either on 0.109 M citrate or CTAD, and plasma was decanted following 20 min centrifugation (2000× *g*), at room temperature (RT), and then stored frozen at <−70 °C until use. Plasmas were thawed for 5 min in a water bath at 37 °C just before use.

### 2.2. Heparin Anti-FXa Kinetic Chromogenic Assays

Heparin anti-FXa kinetic chromogenic assays were obtained from various manufacturers: STA-Liquid Anti-Xa (reagent A), STA-Multi-Hep Calibrator, STA-Quality HNF/UFH, and STA-Quality HBPM/LMWH from Diagnostica Stago (Asnières, France); HemosIL Liquid Anti-Xa (reagent B), HemosIL Heparin Calibrators, HemosIL UF Heparin Controls, and HemosIL LMW Heparin Controls from IL-Werfen (Le Pré Saint Gervais, France); INNOVANCE Heparin (reagent C), INNOVANCE Heparin Calibrator, INNOVANCE Heparin UF Controls, and INNOVANCE Heparin LMW Controls from Siemens (Aubervilliers, France); BIOPHEN Heparin LRT (reagent D), the 2-stage assays BIOPHEN Heparin Anti-Xa-2-stages and BIOPHEN Heparin Anti-IIa-2-stages, BIOPHEN UFH calibrator and controls, and BIOPHEN LMWH calibrator and controls, were from HYPHEN BioMed (Neuville sur Oise, France). For each system, calibrators were tested in duplicate. IL-Werfen and Diagnostica Stago propose plasma calibrators prepared by mixing or combining UFH and LMWH with traceability to International Standards and claim a hybrid calibration curve which can be used irrelevantly for UFH or LMWH. Siemens and Hyphen BioMed propose plasma calibrators were prepared with LMWH only, and assay conditions allowing to obtain full superimposition of UFH and LMWH calibration curves; different types of LMWH were used for Siemens and Hyphen BioMed heparin calibrators. Lastly, Siemens, IL-Werfen, and HYPHEN BioMed anti-Xa reagents (reagents B, C and D) contained dextran sulfate (DS), whilst that from Diagnostica Stago (reagent A) did not.

### 2.3. Reference Materials Used for UFH or LMWH

Reference materials used for UFH or LMWH were the WHO International Standards (IS), obtained from the National Institute for Biological Standards and Controls (NIBSC, Potters Bar, UK), IS 11/176 for LMWH (1068 anti-FXa and 342 anti-FIIa IU per ampoule) and IS 07/328 for UFH (2145 IU per ampoule). These ISs were restored as indicated on the product instructions for use, and a stock solution was prepared at exactly 100 International Units (IU)/mL in a 0.05 M Tris, 0.15 M NaCl, 1% BSA buffer, at pH 7.40 (TBSA). This stock solution was used for preparing UFH or LMWH concentration ranges in the Cryocheck plasma pool, from 0 to 1.8 IU/mL: first a twenty-fold concentrated range was prepared in TBSA (0 to 36 IU/mL); then, 50 µL of each stock solution was spiked in 950 µL of Cryocheck citrate plasma pool to obtain UFH or LMWH concentrations in plasma ranging from 0.00 to 1.80 IU/mL. All spiked plasmas had the same matrix, i.e., 95% Cryocheck plasma pool and 5% TBSA.

### 2.4. Laboratory Coagulation Automated Instruments

Each heparin anti-FXa assay was used according to the associated manufacturer’s instrument: Diagnostica Stago reagent A with STA-R Max (Diagnostica Stago, Asnières, France); IL-Werfen reagent B with ACL-Top 550 (IL Werfen, Le Pré St Gervais, France); Siemens reagent C with CS-5100 (an automated instrument from Sysmex, Kobe, Japan, and distributed by Siemens Healthineers, Aubervilliers, France); HYPHEN BioMed reagent D with the Sysmex CS-5100 instrument (Sysmex, Kobe, Japan); the BIOPHEN anti-FIIa and anti-Xa 2-stage assays were used with the CS-2400 instrument (Sysmex, Kobe, Japan). Reagents were used respecting strictly the recommended manufacturers’ protocols. HYPHEN BioMed reagents, proposed for multiplatform applications, were used with the CE marked protocols developed and validated for CS-5100. All assayed plasmas were tested undiluted (reagent B) or diluted, as claimed in the instructions for use for each assay. Plasma diluent was Owren Veronal Buffer (reagents A and C) or 0.15 M sodium chloride (reagent D).

### 2.5. Verification of Dose–Response Curves for UFH and LMWH

The citrate plasma pool supplemented with either UFH or LMWH ISs was assayed in duplicate for each reagent-instrument combination (reagents A, B, C, and D), comparatively to the manufacturers’ calibrators. Each proposed manufacturer’s heparin calibrator and the UFH or LMWH WHO ISs spiked in plasma, for concentrations from 0.00 to 1.80 IU/mL as described before, were tested with each anti-FXa reagent-instrument combination (reagents A, B, C, and D). For each combination, the 3 calibration curves obtained (heparin assay manufacturer’s calibrator, UFH IS, and LMWH IS) were compared.

### 2.6. Comparison Studies

All plasmas, from UFH or LMWH treated patients, and anticoagulated with citrate or CTAD, were tested using the 4 anti-FXa assay combinations and correlation diagrams were established. Then, a subanalysis was performed for the different groups: plasmas from UFH or LMWH treated patients; use of citrate or CTAD anticoagulant and four groups were obtained, i.e., UFH-citrate, UFH-CTAD, LMWH-citrate, and LMWH-CTAD.

### 2.7. Characteristics of Heparin Plasma Calibrators

Heparin calibrators from the various manufacturers were assayed in duplicate with the 2-stage anti-FXa or anti-FIIa assays with the CS-2400 instrument and compared with the calibrations obtained with UFH or LMWH WHO ISs spiked in plasma. This measurement allowed analyzing the content of each plasma calibrator by establishing the anti-FXa/anti-FIIa ratios: UFH has a ratio of 1.00, whilst depending on the branded material LMWH has a ratio from 1.6 to 9.7 [4,43].

### 2.8. Statistics

Statistics were performed using Analyse-it software version 5.11 (Analyse-it Software, Leeds, UK). Pearson’s correlation diagrams and ordinary least square regression analyses were performed. The heparin concentrations measured in patients’ plasmas were compared using the Friedman’s test; to determine whether any of the differences between the medians were statistically significant, we compared the *p*-value at the 1% significance level to assess the null hypothesis. According to sample size and the non-normal distribution of values, Friedman’s test was more appropriate than the ANOVA multivariate analysis for this study.

## 3. Results

### 3.1. Calibration Curves for the Different Assays

The calibration curves obtained with each anti-FXa combination for the manufacturer’s calibrator and the UFH or LMWH WHO ISs are shown in Figure 1. Superimposition between the manufacturer calibration curves and those obtained with the WHO International UFH or LMWH standards are globally good, although some slight deviations can be seen depending on the system used.

In combination reagent A, UFH IS calibration lacks linearity, especially in the low range, and absorbances are above the manufacturer’s calibration, which can result in underestimation of UFH concentrations, especially for low heparin concentrations. Superimposition is better in the high range. The lack of linearity (R^2^ = 0.979) contributes to underestimating UFH concentration, i.e., the measured absorbance, when plotted against the manufacturer’s calibrator gives significantly lower UFH measurements than when plotted against the UFH WHO IS. In combination reagent B, UFH and LMWH IS calibrations have an acceptable superimposition, and the manufacturer’s calibration has absorbances slightly lower than ISs. This can also contribute to underestimating UFH or LMWH concentrations. In combination reagent C, superimposition is acceptable; the assay calibrator behaves similar to UFH IS and is slightly above that of LMWH IS; this low deviation tends to slightly underestimate LMWH; superimposition for all the curves is also obtained for the combination reagent D.

Differences in heparin concentrations measured among the assays are higher for UFH, especially for low concentrations, and when DS is not present in the assay system. A better accuracy and exactness are obtained when heparin plasma calibrator concentrations are regularly distributed over the dynamic range rather than concentrated in the lower part, as for combination reagent B.

### 3.2. Correlation Studies

Correlation studies are performed on the global group, using plasmas obtained from UFH or LMWH treated patients and anticoagulated with citrate or CTAD. Figure 2 shows the correlation diagrams, for the cross-comparison of anti-FXa manufacturers’ devices: reagents D vs. B, reagents D vs. C, reagents B vs. C, reagents D vs. A, reagents C vs. A, and reagents B vs. A. The ordinary least square fits and correlation coefficients are shown on each graph. The reagents containing DS (reagents B, C, and D) present global acceptable correlations between them on the full assay range, whilst differences tend to be higher when these reagents are compared with reagent A, designed without DS, especially in the low range. This is incompletely reflected by the Pearson’s correlation coefficients calculated on the full dynamic assay range (Figure 1), as r values are always >0.90. Then, a more detailed analysis is necessary.

Differences between assays are higher for UFH samples than for LMWH. The correlation line tendency for reagents A and B shows that heparin concentrations tend to be underestimates, as compared with reagents C and D, and as expected from the calibration curves’ analysis.

The mean values for the different subgroups of plasmas tested (UFH or LMWH), and anticoagulated with citrate or CTAD, are shown in Table 1. The statistical analysis with the Friedman’s test shows that at a significance level of 1%, measured concentrations of citrate anticoagulated samples differ: for A compared with C and D, but not B; for B compared with C and D, but not A; for C compared with A and B, but not D; for D compared with A and B, but not C. Table 2 presents the *p*-values for the citrate or CTAD subgroups. When the CTAD anticoagulated plasmas were tested, the significant differences remained only for: reagents A with C; reagents B with C and D; reagents C with A and B; reagents D with B. Mean heparin concentrations are lower when measured with reagents A and B than with reagents C and D. Differences result partly from the presence of dextran sulfate in the assay formulation, and partly from the manufacturers’ heparin calibrator deviations from UFH or LMWH ISs. Interestingly, no significant differences are noted for the citrate or the CTAD groups between assays for reagents A and B, designed with a combination of UFH and LMWH for calibrators, and between reagents C and D, designed with the use of only LMWH.

### 3.3. Impact of Anticoagulant

To understand and illustrate which major factors are responsible for heparin concentration differences between assays, the correlation diagrams were drawn by identifying each patient’s plasma group. Figure 3 and Table 3 show, for each combination, the correlation diagrams with a separate identification of each subgroup: UFH-citrate, UFH-CTAD, LMWH-citrate, and LMWH-CTAD. These diagrams clearly show that the differences are mainly due to UFH-citrate, and to a lesser extent LMWH-citrate. When CTAD is used as an anticoagulant, a much better coherence of heparin concentrations measured is obtained for all assays.

To confirm the factors explaining the heparin concentration differences measured with the different reagents, especially when designed with or without DS, correlations were analyzed separately for each group of plasma samples. Figure 4 presents the correlation diagrams for UFH or LMWH plasmas anticoagulated either with citrate or CTAD, for the comparison of reagents A and D. Similar correlations are obtained for reagent A as compared with reagents B or C (data not shown).

The highest dispersion of results between reagents A and D concerns UFH samples collected with the citrate anticoagulant. When the same samples are collected with the CTAD anticoagulant, a much better correlation is obtained which was also the case for reagent A as compared with reagents B or C, whilst correlations were better when reagents B, C, and D were compared (*r* > 0.95).

These data suggest that UFH is partially inhibited ex vivo by heparin neutralizing proteins, and its concentration is underestimated when reagent A is used. The presence of DS prevents this inhibition.

The mean heparin concentrations measured with the four anti-FXa assays’ combinations were analyzed for each of the subgroups treated with either UFH or LMWH, and anticoagulated with citrate or CTAD. Table 4 shows the values obtained for each subgroup, underlining the important impact of the anticoagulant used for blood collection, and of the assay design without DS, on the heparin concentrations measured, especially for the low concentration range. Other differences observed with the different assays and the different groups can be explained by the calibration curves biases, when compared with the UFH or LMWH ISs reference curves. An additional impact is noted for reagent B in the low UFH range, and in a lesser extent for reagent C, as the manufacturers’ calibration curves deviate slightly from ISs for UFH or LMWH, as shown in Figure 1.

### 3.4. Composition of the Various Heparin Calibrators

As heparin anti-FXa reagents are indicated for testing all heparin types, manufacturer proposed superimposed curves or hybrid curves can be used irrelevantly for testing UFH or LMWH with the same heparin calibrator. We evaluated the specific FXa or FIIa anticoagulant activity of each heparin plasma calibrator, with the two-stage assays. The specific anti-FXa to anti-FIIa ratios were calculated for each calibrator. The results are presented in Table 5.

UFH has an anti-FXa/anti-FIIa ratio of 1.00 and the different LMWHs have ratios ranging from 1.6 to 9.7, partly dependent on the MW size distribution, and on the pentasaccharide density. From these data, it can be deduced that the Stago heparin calibrator set contains two calibrators (Calibrators 2 and 4) obtained with UFH and two calibrators obtained with LMWH (Calibrators 3 and 5), whilst all the IL HemosIL heparin calibrators contain a mixture of UFH and LMWH. Siemens and HYPHEN BioMed heparin calibrators are homogenous and prepared with only LMWH. The anti-FXa/anti-FIIa ratio shows that different LMWHs are used: this ratio (mean of 2.10) is lower for the Siemens calibrators, similar to that of certoparin or fragmin, and higher for HYPHEN BioMed (mean of 4.02), similar to that of enoxaparin [43]. The WHO International Standard for LMWH 11/176 has an anti-FXa/FIIa ratio of 3.12 (1068 IU for anti-FXa and 342 IU for anti-FIIa).

The use of a single heparin calibration curve appropriateness for measuring UFH or LMWH depends, first, on the accuracy of the superimposition of curves obtained with the corresponding ISs. Both WHO standards were proposed for the two heparin types, UFH or LMWH, which present different inhibition kinetic characteristics.

## 4. Discussion

Recent reports and ECAT surveys have pointed out the variability of heparin measurements using various commercially available anti-FXa assays [55,56,57]. This debate has been reactivated with the extended use of heparin therapy in COVID-19 patients, with accumulation when drug clearance is decreased, or drug resistance when strong inflammation, NETs, and histones are present [14,15,17,66]. Then, heparin concentrations measured can deviate from expected values. Some studies suggest an overestimation of measured heparin concentrations, especially for UFH, when DS is used for anti-FXa assay formulation, whilst other reports support this technical choice as providing the most accurate estimation of anticoagulant activity [55,59,60]. This debate also questions the appropriate residual heparin concentration following neutralization with protamine sulfate in extra-corporeal circulation, when the rebound effect is observed [59,67,68,69]. Studies using heparinase or heparanase have shown that the measured residual heparin does not always match with the anticoagulant activity measured [70,71], and the presence of DS can provide a more reliable measurement. As developers of heparin measurement methods, we have analyzed these different reports and evaluated the various anti-FXa assays’ performances through our experience. In a recent study, Hollestelle et al. already reported the high variability of UFH and LMWH measurements depending on the anti-FXa assay used, and our study confirmed their results using a different and complementary investigation [57]. Hollestelle et al. analyzed the multicentric and longitudinal results from an external quality survey performed on a few samples spiked with UFH or LMWH and provided to all centers participating in an external quality assessment program. Our study is monocentric and performed in the real-life conditions by testing plasmas from UFH or LMWH treated patients and aims to investigate the causes for discrepancies. We show that the presence of DS is important for assay reactivity, in line with former studies, and the heparin calibrator formulation used for UFH and LMWH is also essential, a better linearity being obtained when only LMWH is used.

In this study, we analyzed the various factors impacting the measurement of UFH or LMWH in plasma using four major commercially available anti-FXa assays. We investigated the incidence of reagents’ formulations and that of heparin calibrators, tested by comparison with the UFH or LMWH WHO International Standards. Three of the anti-FXa assays (reagents B (IL-Werfen), C (Siemens-Innovance), and D (HYPHEN BioMed-Biophen)) contained DS, to mobilize all plasma heparin and to avoid its inhibition [61,62], whilst the fourth one (reagent A (Stago-STA Liquid Anti-Xa)) did not. Tested plasma samples, from UFH or LMWH treated patients, were provided by Beaujon University’ Hospital (Clichy, France) anticoagulated with citrate or CTAD. As these plasmas were the left-over tubes from a clinical study, blood was collected using citrate or CTAD, depending on the patient tested. Nowadays, CTAD tends to be used less and less, as laboratories wish to use only one anticoagulant tube type for standardizing hemostasis testing, although it was developed to increase blood and plasma stability for heparin testing [65,72]. CTAD prevents platelet activation and release of heparin neutralization proteins. The assay-to-assay comparison was performed on the global group including 68 plasmas and analyzed for each subgroup of heparin type and anticoagulant used. Then, four groups were obtained: UFH-citrate (*n* = 17); UFH-CTAD (*n* = 11); LMWH-citrate (*n* = 25); LMWH-CTAD (*n* = 15). Globally, correlations were acceptable on the global group among reagents B, C, and D, and there was a higher dispersion of values and a poorer correlation between reagent A and the three others, especially in the low range. When subgroups were analyzed separately, it was obvious that major deviations were observed for UFH-citrate. A better homogeneity was obtained when samples were collected using CTAD, probably because PF4 was then present at a lower concentration. These results strongly suggest that UFH is partly neutralized ex vivo by platelet released proteins, and that this inhibition is prevented using CTAD anticoagulant. When reagents contain DS, this inhibition does not occur, and heparin concentrations measured match better with those present in blood circulation, as reported by various studies [60,61,62,71]. For reagent D, we confirmed that adding PF4, up to a final concentration of 1.0 µg/mL, had no impact on measured values (data not shown).

When comparing the mean heparin concentrations and standard deviations, the lowest values were obtained with reagents A and B, especially for UFH. Although reagents B, C, and D all contained DS, differences were observed among the mean concentrations measured, especially for reagent B as compared with reagents C and D. The calibration curve used for measuring irrelevantly plasma samples containing UFH or LMWH contributes by explaining these differences. Clinical laboratories need a 24/7 available anti-FXa assay for measuring heparin and monitoring treated patients, an analysis that is often requested in emergencies. The heparin type is not always known, and therefore using a single calibration curve is necessary. This procedure needs to be carefully established and validated. Concerning traceability, the WHO proposes two separate International Standards for UFH or LMWH, and manufacturers need to establish anti-FXa assay conditions to obtain superimposable calibration curves for all heparin types. When the correct conditions are fulfilled, calibration curves obtained with UFH or LMWH in plasma can be used without any difference. The heparin calibrations proposed by the various manufacturers differ significantly. Reagents C and D use a calibration curve prepared with only LMWH spiked in a platelet-depleted plasma pool. However, LMWH, used by both manufacturers for reagents C and D, differs as demonstrated by the anti-FXa/anti-FIIa ratios. LMWH is similar to certoparin or fragmin for reagent C and similar to enoxaparin for reagent D. Conversely, reagents A and B propose a hybrid calibration curve obtained by mixing or combining UFH and LMWH in plasma. The anti-FXa/anti-FIIa ratios suggest that calibrators used for reagent B are obtained using a mixture of UFH and LMWH. In contrast, reagent A uses a combination of plasmas containing UFH (Calibrators 2 and 4) or LMWH (Calibrators 3 and 5), with an anti-FXa/anti-FIIa ratio similar to that of certoparin. These choices impact calibration linearity, for UFH mainly, and hybrid calibration curves are intermediate between UFH and LMWH with the introduction of biases. Then, the UFH concentrations tend to be underestimated, as observed with reagents A and B, especially in the low range, and this underestimation is additive with that resulting from heparin inhibition in the absence of DS (reagent A). Different anti-FXa reagents were used along with each manufacturer’s proposed instrument for reagents A, B, and C, by adhering strictly to the instructions for use. Reagent D is proposed for multiplatform applications, developed for all major instruments available. In this study, reagent D was combined with the Sysmex CS-5100 instrument.

The limitations of our study are that the same plasmas were not collected with both anticoagulants tested, i.e., citrate and CTAD, and the number of samples analyzed in each test. However, the results are significant enough to support our analysis and conclusions. Additional and prospective studies would be useful to better document the anti-FXa assays’ differences.

Lastly, when using homogenous (same manufacturer) reagent-instrument systems, reagent weaknesses can be masked by the assay software adaptation, or by introducing algorithms for optimizing the assay apparent performances. This approach is used for adjusting the intrinsic low anti-FXa activity present in all plasma samples, and it is variable from plasma to plasma. In the absence of heparin, this intrinsic anti-FXa activity can account for 0–0.05 IU/mL in normal plasmas, and more rarely up to 0.10 IU/mL. This background activity is likely due to the anti-FXa activity of TFPI, S protein, Z protein (ZP), and ZP inhibitor (ZPI), or the β-AT form. The anti-FXa heparin assay is an inverse relationship between absorbance change measured at 405 nM, and heparin concentration. Therefore, normal plasmas exhibit anti-FXa activity background. Assay systems can manage this variability by masking that effect and “starting to measure” the change in absorbance only from a threshold value, corresponding to plasmas with the highest anti-FXa intrinsic activity. The apparent heparin concentrations in all plasmas are then 0 IU/mL in the absence of heparin, but low concentrations of heparin, in the range 0–0.10 IU/mL, or even up to 0.15 IU/mL, can be missed, which contributes to the underestimation in the low range. This approach is, of course, not possible when the reagent is a multiplatform application, and no adjustment assay software can be used. Then, heparin concentrations measured in plasma are obtained without any data treatment.

## 5. Conclusions

In this report, we provide evidence that supports the usefulness of dextran sulfate for anti-FXa assays used for measuring plasma concentrations of UFH or LMWH, as shown by the good correlations between all assays, designed with or without dextran sulfate, with CTAD anticoagulated plasmas (platelet activation and release of heparin neutralizing proteins is prevented), but not with citrate anticoagulant. In addition, assay variability can result from the heparin calibration type used, the exactness of the UFH and LMWH calibration curves’ superimposition, and the assay software for treatment of assay raw data. The analyses of these factors help to better understanding the differences reported in many studies for the heparin concentrations measured with different anti-FXa reagents.

## Figures and Tables

**Figure 1 biomedicines-09-00700-f001:**
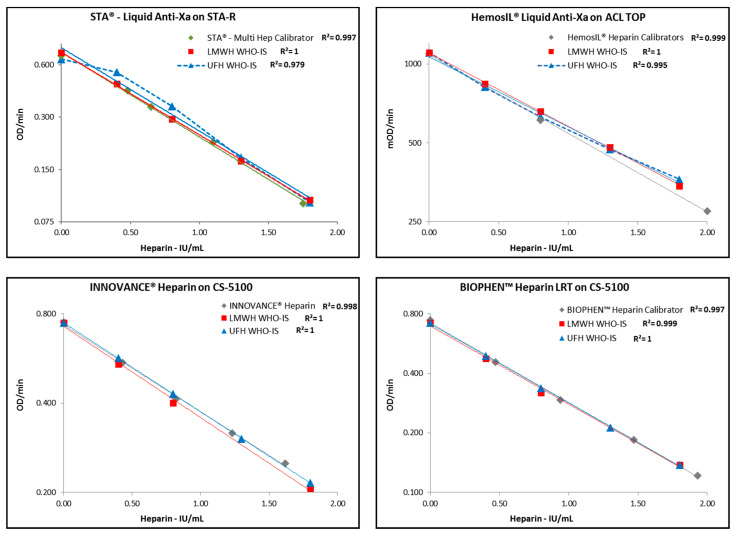
Comparison of calibration curves, tested in duplicate, for each anti-FXa assay used with the manufacturer’s coagulation instrument as compared with the International Standards for UFH or LMWH. Heparin concentrations are on abscissae and change in absorbance per minute (OD/min) on ordinates. Linear regression lines are drawn, and correlation coefficients indicated for each calibration.

**Figure 2 biomedicines-09-00700-f002:**
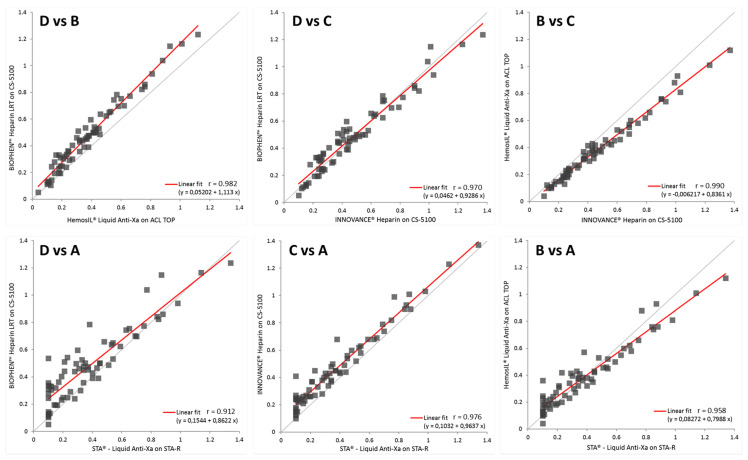
Shows the global Pearson’s cross-correlations among the 4 different branded anti-FXa assays for all the tested plasma samples from heparin treated patients (either with UFH or LMWH), and anticoagulated with citrate or CTAD. Assays for reagents B (IL), C (Siemens), and D (Hyphen BioMed) contain dextran sulfate, whilst the assay for reagent A (Diagnostica Stago) does not. The global ordinary least square correlation coefficient “*r*” is closer to 1.00, when assays containing dextran sulfate are compared between them.

**Figure 3 biomedicines-09-00700-f003:**
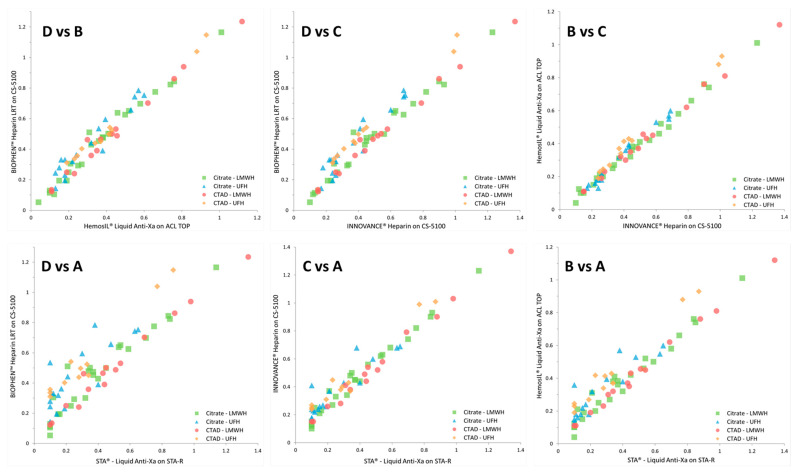
Pearson’s cross-correlations for the comparison of the tested subgroups (UFH-citrate, blue triangles; LMWH-citrate, green squares; UFH-CTAD, orange dots; LMWH-citrate, orange diamonds) with the different reagent-instrument combinations (i.e., reagents A, B, C, and D).

**Figure 4 biomedicines-09-00700-f004:**
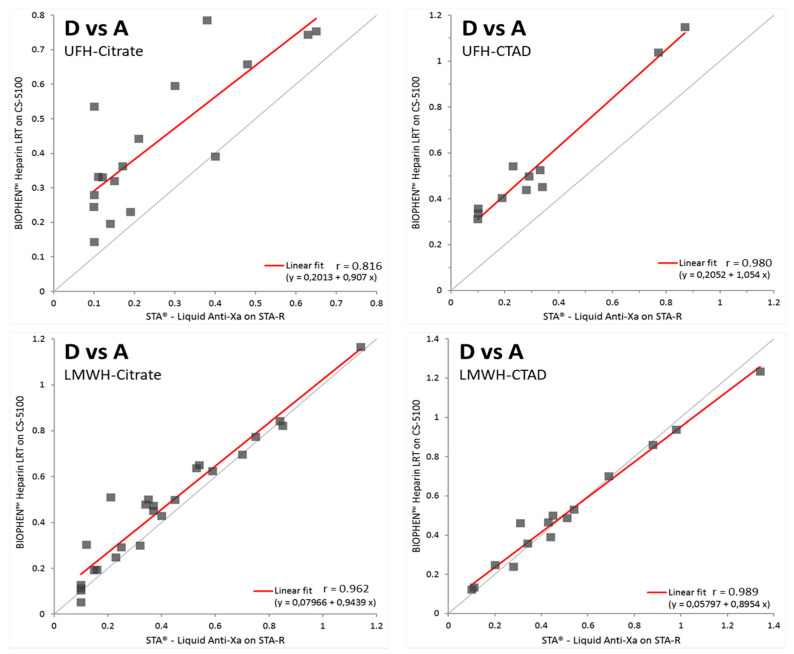
Correlation diagrams between the anti-FXa reagent designed without dextran sulfate (reagent A) and another one with (reagent D) for the different subgroups of tested samples: UFH-citrate, LMWH-citrate, UFH-CTAD, LMWH-CTAD. The correlation is poor for citrate anticoagulated samples, and concentrations are underestimated, especially for UFH, whilst it is acceptable for UFH or LMWH CTAD anticoagulated plasmas, shown by the ordinary least square fit line close to the identity line.

**Table 1 biomedicines-09-00700-t001:** Mean heparin concentrations (IU/mL) measured on the 68 plasmas from UFH or LMWH treated patients, using the 4 different anti-FXa kinetic assays (Stago, Werfen-IL, Siemens, and HYPHEN BioMed), blood was collected either on citrate or CTAD anticoagulant, and plasma decanted following centrifugation.

N = 68	Mean	SD	Minimum	Median	Maximum
	(IU/mL)	(IU/mL)	(IU/mL)	(IU/mL)	(IU/mL)
**STA^®^-Liquid Anti-Xa (A)**	0.376	0.282	0.1	0.315	1.34
**BIOPHEN™ Heparin LRT (B)**	0.479	0.266	0.05	0.452	1.24
**HemosIL^®^ Liquid Anti-Xa (C)**	0.383	0.235	0.04	0.355	1.12
**INNOVANCE^®^ Heparin (D)**	0.466	0.278	0.1	0.41	1.37

**Table 2 biomedicines-09-00700-t002:** *p*-values using the Friedman’s test for cross-comparison of the various assay-tested citrate (*n* = 42) or CTAD (*n* = 26) anticoagulated samples. The median of the populations at the 1% significance level are with *p*-values < 0.01 when the measured concentrations are not similar.

*p* Value	Anti-Fxa Assays	A: STA^®^-Liquid Anti-Xa	B: HemosIL^®^ Liquid Anti-Xa	C: INNOVANCE^®^ Heparin	D: BIOPHEN™ Heparin LRT
**Citrate anticoagulant *n* = 42**	**A** **: STA^®^-Liquid Anti-Xa**	-	0.7456	<0.0001	<0.0001
	**B:** **HemosIL^®^ Liquid Anti-Xa**	0.7456	-	<0.0001	<0.0001
	**C:** **INNOVANCE^®^ Heparin**	<0.0001	<0.0001	-	0.7576
	**D:** **BIOPHEN™ Heparin LRT**	<0.0001	<0.0001	0.7576	-
**CTAD anticoagulant *n* = 26**	**A** **: STA^®^-Liquid Anti-Xa**	-	0.8415	<0.0001	0.0186
	**B:** **HemosIL^®^ Liquid Anti-Xa**	0.8415	-	<0.0001	<0.0001
	**C:** **INNOVANCE^®^ Heparin**	<0.0001	<0.0001	-	0.6949
	**D:** **BIOPHEN™ Heparin LRT**	0.0186	<0.0001	0.6949	-

**Table 3 biomedicines-09-00700-t003:** The Pearson’s correlation coefficients are shown in the tables associated with this Figure 3, for the 4 subgroups: LMWH-citrate (*n* = 25), LMWH-CTAD (*n* = 15), UFH-citrate (*n* = 17), UFH-CTAD (*n* = 11). When *r* > 0.95, correlation between assays looks acceptable, and measurements differ when *r* ≤ 0.95. The highest differences are observed for UFH-citrate for the comparisons between anti-FXa reagents containing dextran sulfate (reagents B, C, and D) with that without (reagent A).

R	Anti-FXa Assays	A: STA^®^-Liquid Anti-Xa	B: HemsIL^®^ Liquid Anti-Xa	C: INNOVANCE^®^ Heparin	D: BIOPHEN™ Heparin LRT
**LMWH-citrate** **(*n* = 25)**	**A: STA^®^-Liquid**	-	0.984	0.99	0.962
	**Anti-Xa**				
	**B: HemosIL^®^ Liquid**	0.984	-	0.996	0.988
	**Anti-Xa**				
	**C: INNOVANCE^®^**	0.99	0.996	-	0.987
	**Heparin**				
	**D: BIOPHEN™**	0.962	0.988	0.987	-
	**Heparin LRT**				
**LMWH-CTAD** **(*n* = 15)**	**A: STA^®^-iquid**	-	0.997	0.995	0.989
	**Anti-Xa**				
	**B: HemosIL^®^ Liquid**	0.997	-	0.997	0.993
	**Anti-Xa**				
	**C: INNOVANCE^®^**	0.995	0.997	-	0.996
	**Heparin**				
	**D: BIOPHEN™**	0.989	0.993	0.996	-
	**Heparin LRT**				
**UFH-citrate** **(*n* = 17)**	**A: STA^®^—Liquid**	-	0.9	0.911	0.816
	**Anti-Xa**				
	**B: HemosIL^®^ Liquid**	0.9	-	0.99	0.962
	**Anti-Xa**				
	**C: INNOVANCE^®^**	0.911	0.99	-	0.955
	**Heparin**				
	**D: BIOPHEN™**	0.816	0.962	0.955	-
	**Heparin LRT**				
**UFH-CTAD** **(*n* = 11)**	**A: STA^®^-Liquid**	-	0.986	0.98	0.98
	**Anti-Xa**				
	**B: HemosIL^®^ Liquid**	0.986	-	0.995	0.996
	**Anti-Xa**				
	**C: INNOVANCE^®^**	0.98	0.995	-	0.997
	**Heparin**				
	**D: BIOPHEN™**	0.98	0.996	0.997	-
	**Heparin LRT**				

**Table 4 biomedicines-09-00700-t004:** Mean heparin concentrations, in IU/mL, measured with the 4 different anti-FXa reagents on the four subgroups: UFH-citrate, LMWH-citrate, UFH-CTAD, and LMWH-CTAD); combinations A and B were used to measure lower concentrations than combinations C and D, although the causes differed (absence of DS in A and combination of UFH and LWWH for calibrators in B).

			A: STA^®^-Liquid Anti-Xa	B: HemosIL^®^ Liquid Anti-Xa	C: INNOVANCE^®^ Heparin	D: BIOPHEN™ Heparin LRT
**UFH (IU/mL)**	**Citrate**	Mean	0.25	0.31	0.37	0.43
	***n* = 17**	Min–Max	0.10–0.65	0.13–0.60	0.17–0.69	0.14–0.79
	**CTAD**	Mean	0.33	0.43	0.46	0.55
	***n* = 11**	Min-Max	0.10–0.87	0.19–0.93	0.24–1.01	0.31–1.15
**LMWH (IU/mL)**	**Citrate**	Mean	0.4	0.38	0.48	0.46
	***n* = 25**	Min–Max	0.10–1.14	0.04–1.01	0.10–1.23	0.05–1.17
	**CTAD**	Mean	0.51	0.44	0.55	0.51
	***n* = 15**	Min–Max	0.10–1.34	0.11–1.12	0.15–1.37	0.12–1.24

**Table 5 biomedicines-09-00700-t005:** Analysis of the various heparin calibrators from the different manufacturers by testing anti-FXa and anti-FIIa activities (IU/mL) as compared with the manufacturers’ claimed heparin concentrations and anti-FXa/anti-FIIa ratios: Diagnostica-Stago has 2 calibrators with UFH and 2 with LMWH; IL uses a combination of UFH and LMWH for each calibrator; Siemens and Hyphen BioMed calibrators are prepared with only LMWH.

Brand	Heparin Calibrators	Anti-IIa IU/mL	Anti-FXa IU/mL	Anti-FXa/Anti-IIa Ratio	Manufacturer’s Target Value IU/mL
**Instrumentation Laboratory**	1	0.00	0.02	0.00	0.00
	2	0.59	0.77	1.32	0.80
	3	1.68	2.32	1.38	2.00
**Diagnostica Stago**	1	0.00	0.02	0.00	0.00
	2	0.59	0.57	0.98	0.40
	3	0.38	0.82	2.08	0.68
	4	1.12	1.08	0.95	0.98
	5	1.06	2.18	2.06	1.79
**Siemens**	1	0.00	0.00	0.00	0.00
	2	0.18	0.37	2.06	0.43
	3	0.38	0.79	2.08	0.85
	4	0.56	1.19	2.13	1.27
	5	0.74	1.58	2.14	1.67
**HYPHEN BioMed**	1	0.00	0.00	0.00	0.00
	2	0.11	0.44	4.00	0.45
	3	0.23	0.92	4.00	0.92
	4	0.33	1.35	4.09	1.36
	5	0.48	1.92	4.00	1.80

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
