# Peer review of "Optimization of Heparin Monitoring with Anti-FXa Assays and the Impact of Dextran Sulfate for Measuring All Drug Activity"

_biomedicines, 2021, doi:10.3390/biomedicines9060700_

Round 1

Reviewer 1 Report

Amiral and colleagues compared four major commercially available anti-FXa assays. They compared the calibration curves of each assay with those obtained using the International Standards for UFH or LMWH, then they evaluated 68 samples from treated patients and compared the obtained results. Subsequently, they studied the impact of the anticoagulant (citrate and CTAD) on the anti-FXa results as well as the composition of the various heparin calibrators. Many concerns warrant further explanation and attention by the authors.

1/- The introduction is too long. Many detailed information is out of topic. It should be reduced at least by 60%.

2/- No statistical analysis is reported in the comparison of the different assays results in between. Why? ANOVA or other statistical tests should be applied, shouldn’t they?

3/- Some heparin concentrations are lower when measured with reagent B. Authors claimed that it was due to the calibration used, while in figure 1B, UFH and LMWH ISs calibrations have good superimposition, particularly for values below 0.6-0.7 IU/mL. This point should be clarified.

4/- Based on Table 2, the difference between the mean values obtained with system A and B, seems less important than that observed between the systems B and D either for UFH or LMWH samples. No statistical analysis was performed. This important issue should be addressed by the authors. Some significant differences might exist between the systems B, C and D even though they all include DS.

5/- Authors claimed that their results strongly suggest that “UFH is partly neutralized ex vivo by platelet released proteins and that this inhibition is prevented using CTAD anticoagulant”. However, there is no data in the present manuscript that are in favor of such hypothesis even if it is highly likely. Blood from each patient was collected either on citrate or CTAD. To verify this hypothesis, it should have been collected on both anticoagulants and anti-FXa should have been measured and compared in between using the same anti-FXa measuring system.

6/- No limitation is mentioned in the discussion section.

7/- The discussion contains a fair amount of repetition of materials and results paragraphs. It should be reviewed and shortened.    

8/- Hollestelle et al. have recently reported in Clin Chem Lab Med 2020 significant differences in measured anti-Xa values between the frequently used methods and that this may be caused by differences in method composition, such as the addition of dextran sulphate. They also published substantial inter-laboratory variation in anti-FXa monitoring, based on External quality assessment (EQA) data from multiple years from more than 100 laboratories, particularly at low UFH and LMWH concentrations. They concluded that the choice of the anti-FXa method is particularly important for UFH and LMWH measurement and that the variation in measurements may have an effect on clinical implications, such as therapeutic ranges. In light of their results, what is the added value of the present study? please discuss and comment.

Author Response

Answers to reviewer 1:

Many thanks to the reviewer for the extensive and accurate analysis of our manuscript and the pertinent comments and suggestions done. This helps a lot in improving the quality and soundness of our report. Accordingly, our article has been extensively revised and modified, and the results analyzed with the use of statistics. The detailed answer to the various reviewer’s concerns are here below:

1/- The introduction is too long. Many detailed information is out of topic. It should be reduced at least by 60%.

Answer: The introduction has been extensively revised, modified and reduced; one paragraph has been withdrawn, and the 3 other paragraphs have been revised and shortened. Our introduction focusses mainly on the background of heparin therapy, the usefulness of the anti-FXa methods for measuring drug concentration, and on the various recent articles keeping the attention on the assay variability, especially for UFH, but also for LMWH.

2/- No statistical analysis is reported in the comparison of the different assays results in between. Why? ANOVA or other statistical tests should be applied, shouldn’t they?

Answer: Statistics have been introduced for analysis of results, as suggested. Our study is performed on clinical samples from hospitalized patients treated with UFH or LMWH. The distribution of heparin concentrations is not a normal distribution in our study group, and therefore the ANOVA multivariate analysis is not appropriate (of course with the ANOVA multivariate analysis all assays are found different between them, whether the comparison between assays is, with a p-value <0.0001). We preferred to use the Friedman’s test to establish the p-value for the median concentrations. This test allows to illustrate which assays measure similar concentrations and which ones differ. We also used the Pearson’s correlation diagrams and the ordinary least-square regression analysis, as shown on figure 3. Statistics used are now described in materials and methods (paragraph 2.8).

3/- Some heparin concentrations are lower when measured with reagent B. Authors claimed that it was due to the calibration used, while in figure 1B, UFH and LMWH ISs calibrations have good superimposition, particularly for values below 0.6-0.7 IU/mL. This point should be clarified.

Answer: The concentrations measured with reagent B are lower when compared to those measured with reagents C or D, although dextran sulfate is present in all these assays. The reason is the calibration used, which is obtained with a mixture of UFH and LMWH. Calibrations with IS for UFH or LMWH are acceptable for reagent B, with r² > 0.99, but the manufacturers heparin calibration curve is slightly below those obtained with ISs. This can contribute to lower the measured concentrations.

4/- Based on Table 2, the difference between the mean values obtained with system A and B, seems less important than that observed between the systems B and D either for UFH or LMWH samples. No statistical analysis was performed. This important issue should be addressed by the authors. Some significant differences might exist between the systems B, C and D even though they all include DS.

Answer: As explained now in results and discussed in the discussion section, there are 3 factors which contribute to differences between assays: i) presence of dextran sulfate, which prevents from heparin neutralization by platelet released proteins, mainly PF4; ii) the formulation of heparin calibrators, which offer a better linearity and accuracy when obtained with a single heparin type; iii) the superimposition of UFH and LMWH calibration curves performed with ISs, and the compliance of manufacturer’ assay calibrator with these curves. This is analyzed in results and discussed in the discussion section.

5/- Authors claimed that their results strongly suggest that “UFH is partly neutralized ex vivo by platelet released proteins and that this inhibition is prevented using CTAD anticoagulant”. However, there is no data in the present manuscript that are in favor of such hypothesis even if it is highly likely. Blood from each patient was collected either on citrate or CTAD. To verify this hypothesis, it should have been collected on both anticoagulants and anti-FXa should have been measured and compared in between using the same anti-FXa measuring system.

Answer: The reviewer is right on that a limitation of our study relies on the use of either citrate or CTAD anticoagulant for our study. As we used left-over plasma tubes from a clinical study, with some patients collected on citrate and others on CTAD, plasmas from the same patients with both anticoagulants were not available. However, as shown on figure 4 for the comparison between reagents A (without dextran sulfate) and D (with dextran sulfate) on citrate or CTAD anticoagulated plasmas, it is obvious that in the absence of dextran sulfate some heparin is neutralized when plasma is anticoagulated with citrate but not when CTAD is used, as shown by the correlation and identity lines: parallel but separate for citrate, and very close for CTAD. We also show the p-values for the Friedman’s test on the citrate and CTAD groups on the table associated with figure 3. Lastly, we tested the impact of PF4 spiked in plasma and tested with reagent D, and no interference was observed up to 1.0 µg/ml, which is a very high concentration comparatively to the usual PF4 concentrations in citrated plasma (usually < 0.1 µg/ml, and rarely > 0.5 µg/ml in patients just after the start of heparin therapy).

6/- No limitation is mentioned in the discussion section.

Answer: The study limitations have now been introduced in the discussion section.

7/- The discussion contains a fair amount of repetition of materials and results paragraphs. It should be reviewed and shortened.    

Answer: The discussion has been extensively revised and adjusted to the reviewers’ comments, to focus mainly on factors which explain the differences between assays, and on the analysis comparatively to other articles reporting differences between anti-FXa assays.

8/- Hollestelle et al. have recently reported in Clin Chem Lab Med 2020 significant differences in measured anti-Xa values between the frequently used methods and that this may be caused by differences in method composition, such as the addition of dextran sulphate. They also published substantial inter-laboratory variation in anti-FXa monitoring, based on External quality assessment (EQA) data from multiple years from more than 100 laboratories, particularly at low UFH and LMWH concentrations. They concluded that the choice of the anti-FXa method is particularly important for UFH and LMWH measurement and that the variation in measurements may have an effect on clinical implications, such as therapeutic ranges. In light of their results, what is the added value of the present study? please discuss and comment.

Answer: The article of Hollestelle et al. is now referenced in our report (reference 67), and is mentioned in introduction, then discussed I discussion respectively to our results. The Hollestelle’s investigation concerned a multicentric and longitudinal study on anti-FXa assays, based on the reporting of the external quality assessment values measured locally on few samples spiked with UFH or LMWH (and Fondaparinux or Sodium Danaparoid, but this is not the matter of our paper). In our study, we analyzed plasmas from hospitalized patients and treated either with UFH or LMWH, and collected on citrate or CTAD. We indeed confirm the differences already reported by Hollestelle et al. Both studies are then complementary. In addition, we focused on explaining what could induce these differences and we propose the 3 impacting factors as supported by our results: presence of dextran sulfate; calibrator formulation; assay conditions permitting the full superimposition of UFH and LMWH International Standards calibration curves, and compliance of manufacturers’ calibrators with these standards. The added value of our study is an observational analysis on assay differences, by testing material from the real-life, and not spiked plasmas used for quality surveys, and we bring evidence on the factors responsible for this variability. We hope it can contribute to improve the assay-to-assay standardization.

Reviewer 2 Report

The manuscript reports on the critical comparison of the results of heparin measurements obtained with four commercially available chromogenic assays, by testing many plasma samples collected with two different anticoagulants, citrate and CTDA. The UFH and LMWH WHO International Standards as reference materials in addition to manufacturer’s calibrators were also considered.

The authors provide a sound critical analysis of the various factors influencing the accuracy of quantitative response of the chromogenic assays.

The manuscript is clear, well-argued and with an appropriate bibliographic documentation. It deserves publication after minor corrections are made.

Page 6, line 4: maybe “…slightly overestimate LMWH”

Page 11, line 7: table 3

Page 11 lines 16 and 17: provide reference for certoparin and enoxaparin a-FXa/a-FIIa ratio

Page 12, line 12: maybe “heparanase”?

Page 16, References 72 and 73 are the same

Improve resolution of figures 2 and 3.

Author Response

Many thanks to the reviewer for the useful comments and suggestions, which have been duly considered for modifying the article.

The changes introduced are described here below:

Slightly overestimates LMWH corrected.

Reference to Table 3 corrected.

Refernce for certoparin, fragmin, enoxaparin anti-Xa/anti-IIa ratios has been introduced (reference 43).

Heparanase has been corrected.

Reference 73, the same than 72, has been deleted and reference numbers adjusted.

Figires 2 and 3 have been drawn again, and better documented.

Round 2

Reviewer 1 Report

The authors should be congratulated on their changes in response to many of the points I had raised, most of them being fully appropriate. The manuscript was significantly improved. However, some additional concerns warrant further attention.

1/- Even though the introduction was shortened, it remains too long and many details are out-of-topic and render it somewhat cumbersome. It should be reduced further (at least by 30-40%).

2/- Page 2, L 35: “The assay conditions assays”, please modify

3/- Page 5, L 10: “calculating the median concentrations for the p-value”: please clarify

4/- Figure 1: how many times the calibrations were performed? Could the author report data on the graphs as mean +/- SD or median +/- interquartile ranges?

5/- Figure 1: STA-MultiHep Calibrator seems to underestimate UFH over the range 0.25 and 1.00 IU/mL (therapeutic range), therefore not only in the low range as reported Page 6, L5. Please clarify.

6/- Figure 1: combination B, manufacturer’s calibration seems to deviate below ISs curves for the values above 0.8 IU/mL? Please add precision

7/- Page 6, L 26-27 and Figure 2: “there is a higher dispersion when these reagents are compared with reagent A”: however, C vs A correlation coefficient (0.976) is comparable to that of D vs C. Please comment

8/- Figure 3: the r values reported are already mentioned in Figure 2. It would be more interesting to report the r values that correspond to the 4 subgroups for each pair-wise combination comparison, wouldn’t it?

9/- Page 9, L16-17: “correlations are acceptable when reagents …”. What is the threshold above which correlations were considered as such? It is to mention in the Statistics section of the manuscript.

10/- Figure 4: please add the corresponding r values.

Author Response

Many thanks for this complementary review, the comments to changes, and the additional concerns added. We have worked hard to consider all the reviwer's recommendations and we introduced the changes requested. The corrected version is here attached, with the changes underlined in yellow. We hope that the reviewer will now consider that we duly integrated all the requested modificcations, and that the article is acceptable .

Here below are the detailed answers to the various concerns:

  1. The introduction has again been extensively revised and condensed. The 3 chapters have been condensed in a single one. This introduction is now only focused on the assays' performances comparisons and analysis.
  2.  The "assay conditions" has been modified.
  3. The presentation of  the statistics performed with the Pearson's method has been totally reworded and changed. How the p-value is obtained is defined in materials and methods.
  4. The various calibrations have been performed in duplicate. This study has been conducted in the real-life conditions for testing the clinical samples (not spiked samples) with the recommended manufacturers' protocols, and according to the instructions for use. The results obtained are in full compliance with the various reports showing the differences between assays (mainly undersestimation of UFH by some assays), and especially with the Hollestelle study. Important deviations are observed for UFH, in the low range, when dextran sulfate is not included in the reagent formulation, and more especially when citrate is used as anticoagulant.  Our goal was to analyse the root causes inducing the differences between assays. Performing the calibration curves in duplicate was then appropriate for the assay goal. No standard deviations need then to be reported. We performed this study in the standard working conditions for reagent to reagent comparison.
  5. The global underestimation of UFH by reagent A has been outpointed. In addition to calibrators, the major cause results from the probable UFH neutralization by PF4 and plasma heparin inhibiting proteins, as demonstrated later by the comparison on citrate or CTAD anticoagulated samples.
  6. This point has been better described in the text: the manufacturers' calibration curve is "below" those from International standards for UFH or LMWH. This induces a direct understimation (please, note that in former ECAT reports, when IL was using a single heparin/LMWH for calibration, and not a mixture of UFH and LMWH, or when the Coamatic heparin assay was used, less differences were obtained between reagents B and D, than those present in the most recent reports).
  7. The analysis of differences between the  reagents has been revised. The differences are mainly observed between reagents designed with dextran sulfate (B, C, D) and that without (A), on UFH-citrate samples. The specific p-values are shown on tables associated with figure 3. The subgroup analysis better shows the differences between reagents, especially for the likely interference of heparin neutralization proteins.
  8. Tables with the r-values have been added under figure 3.
  9. We considered that correlations with a r-value < 0.95 show differences between reagents.
  10. The correspondiong r-values have been added on figure 4. 
